# Impact of COVID-19 on A1c Management and Telehealth Use Among a Type 2 Diabetes Mellitus Population in the Outpatient Setting

**DOI:** 10.3390/healthcare13182372

**Published:** 2025-09-22

**Authors:** Megan Jodray, Annesha White, Kimberly G. Fulda, Haley McKeefer, Fan Zhang, Chinemerem Opara, Yan Xiao

**Affiliations:** 1College of Pharmacy, University of North Texas Health Science Center, 3500 Camp Bowie Blvd, Fort Worth, TX 76107, USA; annesha.white@unthsc.edu (A.W.); haleymckeefer@my.unthsc.edu (H.M.); opara@my.unthsc.edu (C.O.); 2Texas College of Osteopathic Medicine, University of North Texas Health Science Center, 3500 Camp Bowie Blvd, Fort Worth, TX 76107, USA; kimberly.fulda@unthsc.edu (K.G.F.); fan.zhang@unthsc.edu (F.Z.); 3College of Nursing and Health Innovation, University of Texas at Arlington, 701 S Nedderman Drive, Arlington, TX 76019, USA; yan.xiao@uta.edu

**Keywords:** COVID-19, type 2 diabetes, hemoglobin A1c, telehealth, diabetes management

## Abstract

Background: The COVID-19 pandemic disrupted routine healthcare delivery, raising concerns about chronic disease management, particularly for individuals with type 2 diabetes mellitus (T2DM). This study evaluated the pandemic’s impact on glycemic control and telehealth utilization in an underserved outpatient population. Methods: A retrospective cohort analysis used de-identified electronic health record and claims data from a family medicine clinic in central Texas. The study included 387 adults with T2DM who had at least one A1c measurement in both the pre-pandemic period (1 March 2019–13 March 2020) and COVID-19 era (14 March 2020–31 March 2021). Outcomes included A1c control (<8.0%), prescription trends, and telehealth use. A case series examined individual-level patterns. Results: A significantly higher percentage of patients achieved A1c control during the COVID-19 era (75.2%) compared to the pre-pandemic period (68.7%, *p* < 0.05), despite a decline in prescriptions for diabetes medications and supplies. Telehealth visits increased substantially. Patients who maintained or improved glycemic control often had uninterrupted access to medications and telehealth. Conclusion: This study is novel in its focus on a safety-net outpatient clinic serving a predominantly low-income, diverse population in central Texas, an underserved group often underrepresented in diabetes research. By combining a retrospective cohort analysis with a descriptive case series, the study offers both population-level trends and individual-level insights into how medication access and telehealth engagement influenced glycemic control during the pandemic. These findings highlight the potential of telehealth to support diabetes management during healthcare disruptions and underscore the importance of maintaining medication access in vulnerable populations.

## 1. Introduction

On 11 March 2020, COVID-19 was declared a pandemic by the World Health Organization (WHO). The announcement came after millions of people were told to wear masks, remain indoors, and avoid hospitals in 2020. The year 2020 included widespread restrictions on seeking adequate care due to lock-downs as well as a shift to increased use of telehealth [1,2]. These restrictions may have impacted patients with diabetes as lack of access for regular check-ups can result in reduced management of the condition, leading to poor health outcomes.

Emerging evidence has shown that COVID-19 not only complicates the management of diabetes but may also directly contribute to its onset and progression, with studies reporting an increased incidence of type 2 diabetes following COVID-19 infection [3]. SARS-CoV-2, the virus responsible for COVID-19, has been shown to damage insulin-producing beta (β) cells in the pancreas, likely through immune-mediated inflammation involving macrophages [4]. This damage can impair insulin secretion, potentially leading to new-onset diabetes or worsening glycemic control in those with pre-existing disease. Additionally, the clinical management of diabetes in patients hospitalized with COVID-19 has highlighted the importance of glucose-lowering therapies in improving outcomes [5]. These findings highlight the need for vigilant glycemic monitoring during and after COVID-19 infection due to both direct and indirect effects on glucose metabolism.

While several studies have explored the relationship between COVID-19 and diabetes severity [6], few have examined how the pandemic affected glycemic control, particularly in underserved populations receiving care in safety-net outpatient settings [7,8]. These populations often face compounded barriers to care, including limited access to medications, supplies, and follow-up services.

Access to healthcare for persons living with chronic disease states is essential to reaching desired therapeutic outcomes and lowering cost. Telehealth has been promoted, especially in low-income and racially diverse populations [2,9], as a method to increase access to care and reduce disparities in specialty care access among patients with diabetes, particularly during the COVID-19 pandemic [10,11]. Telehealth is defined as providing health services from a distance by utilizing telecommunication technology between the healthcare provider and patient. Outcomes, such as improved A1c and reduced disease-related hospitalizations, are reached with the assistance of pharmacotherapy. Managing medication adherence becomes important in reaching those outcomes, and there are different factors that may be evaluated to determine adherence. Adherence may include visiting the doctor regularly, picking up medications when provided a prescription, or administering a medication with the correct dose at the right time [12]. Lack of access to healthcare leads to non-adherence, and non-adherence leads to poor clinical outcomes in patients managing diabetes mellitus [13,14]. One proposed strategy to address access issues among patients was telehealth.

With the use of telehealth, reducing the barriers to accessing healthcare may have prevented a decline in the clinical outcomes in this population. Despite the promotion of the use of telehealth services, access barriers to insulin, other diabetic therapeutics, and diabetic supplies in the outpatient setting may still exist, resulting in poor hemoglobin A1c control for patients with type 2 diabetes.

This study aimed to evaluate the impact of the COVID-19 pandemic on diabetes management in an outpatient setting. The primary objective was to assess changes in hemoglobin A1c control. Secondary objectives included examining trends in prescription access for diabetes medications and supplies and describing the utilization of telehealth services. While telehealth use was quantified, its direct impact on A1c control was not evaluated and remains an area for future research.

By combining a retrospective cohort analysis with a case series of patients whose A1c status changed, this study provides both population-level trends and individual-level insights into diabetes care continuity during a public health emergency. It addresses a critical gap in the literature by focusing on glycemic control and medication access in a low-resource outpatient setting. This study contributes to the literature in several important ways. First, it focuses on an underserved population receiving care in a safety-net outpatient clinic, an area that remains underexplored in existing research. Second, the mixed-methods design, combining retrospective cohort analysis with a descriptive case series, allows for a nuanced understanding of both systemic trends and individual patient experiences. Third, the study reveals a counterintuitive finding: despite a decline in prescriptions for diabetes medications and supplies, glycemic control improved significantly. Finally, while many studies have documented increased telehealth use, our analysis links this trend to specific medication access patterns and glycemic outcomes, highlighting the potential of telehealth to mitigate care disruptions in vulnerable populations.

## 2. Materials and Methods

This study employed a mixed-methods design, combining a retrospective cohort analysis with a descriptive case series. The cohort analysis compared diabetes management outcomes, including hemoglobin A1c (A1c) control, prescription trends, and telehealth utilization, between the pre-COVID-19 period (1 March 2019–13 March 2020) and the COVID-19 era (14 March 2020–31 March 2021). The case series provided additional insight into individual-level care patterns by examining patients whose A1c control status changed between the two periods, with a focus on medication use and telehealth engagement.

### 2.1. Participants

Eligible participants were adults (≥18 years) with a diagnosis of type 2 diabetes who received care at a single academic family medicine clinic in central Texas. Inclusion required at least one recorded A1c measurement in both the pre-COVID-19 and COVID-19 era periods. All patients had continuous access to the clinic’s services throughout the study, which may have contributed to the observed stability in A1c levels across cohorts.

A total of 3102 patients were screened for eligibility based on electronic health record data. Of these, 387 patients met the inclusion criteria, having a diagnosis of type 2 diabetes and at least one recorded A1c measurement in both the pre-COVID-19 and COVID-19 era periods, and were included in the final analysis.

### 2.2. Data Collection

The following data were extracted: demographics (sex, race, ethnicity, insurance, and preferred language), hemoglobin A1c values with dates, dates of visit and visit modality (in-person versus telehealth), and medication prescriptions. Medications from the claims data were categorized based on answering ‘Yes’ or ‘No’ to three items: “Diabetes Medication”, “Diabetes Supply”, or “Insulin.” The dataset did not include information on potential confounding factors such as changes in diet, physical activity, stress levels, or socioeconomic status. These variables were therefore not controlled for in the analysis and represent important limitations.

Prescription data were analyzed as aggregate counts of medications, supplies, and insulin prescriptions. These counts reflect the total number of prescriptions issued during each study period and do not represent the number of unique patients receiving prescriptions. As such, multiple prescriptions for a single patient may be included in the totals.

### 2.3. Outcome Measures

Consistent with the American College of Physicians’ guidance for outpatient diabetes management, A1c control was defined as a hemoglobin A1c level of <8.0% (64 mmol/mol) [15]. This threshold was used to assess glycemic control across both study periods. A1c values were compared between two cohorts, a pre-COVID-19 cohort and a COVID-19 era cohort. Telehealth visits were defined as any outpatient encounter coded as a virtual or telephone visit in the electronic health record. The “virtual” category included both phone and video calls. However, the dataset did not differentiate between these modalities, limiting our ability to assess whether the type of telehealth visit influenced glycemic outcomes. The type of provider (e.g., physician, nurse practitioner, pharmacist) was not differentiated in the analysis and may have varied across visits.

### 2.4. Case Series Analysis

A change was defined as a shift in the last recorded A1c value from >8% to ≤8%, or from ≤8% to >8% (64 mmol/mol). For each of these patients, medication use during the COVID-19 era was categorized into one of three groups: (1) Use of both insulin and other diabetes medications, (2) Use of non-insulin diabetes medications only, or (3) Use of insulin only. Insulin discontinuation was inferred from prescription gaps. A prescription was considered “lost” if it ended without a subsequent prescription for the same drug or drug class within the same therapeutic category, resulting in a gap in therapy. Refill data were not available, so continued use of previously dispensed medications could not be confirmed.

For illustrative purposes, 10 patient cases were randomly selected from the subset of individuals whose A1c status changed between the two study periods. Five patients with improved A1c and five with worsened A1c were chosen using simple random sampling to highlight patterns in medication use and telehealth engagement.

### 2.5. Statistical Analysis

Descriptive statistics were used to summarize patient demographics and clinical characteristics. A Pearson’s Chi-squared test was conducted to assess the association between A1c control status (A1c > 8%) and the study period (pre-COVID-19 vs. COVID-19 era). Statistical significance was set at *p* < 0.05. Analyses were conducted using IBM SPSS Statistics Version 21.0 [16]. No regression analyses were conducted; however, future research may consider regression modeling to adjust for potential confounding variables such as age, race, and gender. The case series analysis provided additional qualitative insights into individual patient experiences and medication use patterns that may not be captured through quantitative methods. All patients with a change in A1c status were evaluated for medication profiles and telehealth engagement.

## 3. Results

### 3.1. Study Design

The study design and methodology, including cohort definitions and case series criteria, are described in detail in Section 2. This section presents the key findings from the retrospective cohort analysis and the descriptive case series.

### 3.2. Outcomes of the Study

As defined in the Methods section, A1c control was considered <8.0% (64 mmol/mol). This threshold was used to compare glycemic control between the pre-COVID-19 and COVID-19 era cohorts.

### 3.3. Definitions and Sample Characteristics

Patient characteristics, including demographics, A1c values, visit modality, and medication prescriptions, were summarized to describe the study population. Detailed data collection methods are described in Section 2.2.

Among 3102 patients who visited the study health center, a total of 387 individuals met the study criteria; 233 (60.2%) were women, white (53.2%), and spoke English as their primary language (85.5%). All patients had some form of health insurance (Table 1).

From the pre-COVID-19 to the COVID-19 era, there was a 9.3% increase in the number of patients reaching their A1c goal of under 8.0% (64 mmol/mol) (*p* < 0.05). The A1c was calculated using the last recorded A1c number in the defined period. There was a statistically significant difference in the mean A1c from pre-COVID-19 cohort compared to COVID-19 era cohort (*p* < 0.05), seen in Table 2 below.

From the pre-COVID-19 to the COVID-19 era, the largest loss occurred in the number of diabetes medication prescriptions by 19.4% (*p* < 0.05), seen in Table 3. No significant changes were observed for the decrease in the number of prescriptions for supplies by 8.9% (*p* = 0.22) or insulin by 11.7% (*p* = 0.13). However, the utilization of telehealth services increased significantly (*p* < 0.05).

Case series analysis identified 81 patients with A1c control status changes. Among the 28 patients whose A1c worsened (i.e., A1c increased from <8.0% to ≥8% [64 mmol/mol], 10 had insulin prescriptions. In contrast, of the 53 patients whose A1c improved (i.e., decreased from ≥8.0% to <8%), 26 did not have insulin prescriptions.

To illustrate these findings, Table 4 presents medication profiles for five patient cases, chosen at random, whose A1c worsened from the pre-COVID-19 to the COVID-19 era. The majority were using insulin throughout the COVID-19 era and experienced changes to their in their non-insulin diabetes medications. A prescription was considered “lost” if it ended without a subsequent prescription for the same drug or drug class, resulting in a gap in therapy.

Most medication changes, including new starts or dose adjustments, occurred during the pre-COVID-19 period. Few prescriptions were initiated during the COVID-19 era, and only a small number continued uninterrupted across both periods.

Table 4 also includes five randomly selected patients whose A1c improved during the study period. Most prescriptions for these patients began in the pre-COVID-19 period and continued into the COVID-19 era. Three of the five patients were on insulin during both periods. While one patient had only three telehealth visits, the others had at least six, all of which were attended. The high number of visits may reflect care for comorbid conditions in addition to diabetes.

## 4. Discussion

This study found that despite a reduction in prescriptions for diabetes medications and supplies during the COVID-19 era, a greater proportion of patients achieved A1c control (<8.0%) compared to the pre-COVID-19 period. These findings are consistent with prior research, such as Ludwig et al. (2021), which reported improved glycemic control during lockdowns, potentially due to increased time for self-care, more consistent routines, and heightened health awareness [17]. However, other studies have reported mixed outcomes depending on the population and healthcare setting [18,19].

Under typical circumstances, an A1c > 8% would prompt a provider to adjust therapy. However, several factors may explain the observed decline in prescriptions. Patients may have adapted by rationing medications, or through improved self-care behaviors [20], using previously stored supplies, or improving lifestyle behaviors such as diet and physical activity. Similar trends were reported by Yunusa et al. (2021), who observed a decline in insulin prescribing during the pandemic [12].

To better understand these patterns, we examined patient case profiles. Patients with improved A1c generally maintained consistent access to medications across both time periods, with most prescriptions originating in the pre-COVID-19 era and continuing through the pandemic. In contrast, patients with worsening A1c often experienced therapy interruptions, suggesting that continuity of medication access played a critical role in glycemic outcomes.

Interestingly, not all patients with A1c > 10% were prescribed insulin. This may reflect clinical decisions to prioritize first-line therapies and lifestyle interventions, particularly in newly diagnosed patients. Additionally, delays in obtaining A1c tests or disruptions in electronic prescription systems may have limited timely therapy adjustments. These findings highlight the need to further explore barriers to prescription access during public health emergencies.

Our study builds upon prior research, including the work by Papachristoforou et al. (2022), which examined diabetes management during the COVID-19 pandemic [21]. However, our study is distinct in several ways. It focuses on a safety-net outpatient clinic serving a predominantly low-income, racially diverse population in central Texas, an underserved group often underrepresented in diabetes research. Additionally, our mixed-methods design, combining a retrospective cohort analysis with a descriptive case series, provides both population-level trends and individual-level insights. Notably, we observed a counterintuitive improvement in A1c control despite a decline in prescriptions, suggesting adaptive behaviors or unmeasured support mechanisms. Furthermore, our analysis links increased telehealth utilization to medication access patterns and glycemic outcomes, underscoring the potential of telehealth to mitigate care disruptions in vulnerable populations.

The stability in average A1c levels, despite decreased prescription volume, suggests that patients may have adapted by rationing medications, or through improved self-care behaviors [20], using previously stored supplies, or improving lifestyle behaviors such as diet and physical activity. That said, our study did not include behavioral data, such as changes in diet, exercise, or stress, which limits our ability to fully explain the counterintuitive improvement in glycemic control observed during the pandemic. To better understand the mechanisms driving these changes, future research should incorporate pharmacy fill data, patient-reported adherence, and behavioral metrics.

Building on this context, our dataset, collected after the WHO declared COVID-19 a pandemic and before widespread vaccine availability, provides a unique window into how patients managed diabetes during a period of heightened healthcare disruption. We hypothesized that quarantine-related challenges, such as increased sedentary behavior and reduced access to prescriptions, would negatively impact diabetes management. Indeed, prescription volumes declined across all categories, though the decreases were modest (<20%). Nevertheless, even small disruptions in access to medications or supplies (e.g., glucometer strips) could impair patients’ ability to monitor and manage their blood glucose effectively.

This study employed descriptive statistics and chi-squared tests to assess associations between A1c control and study period. However, we did not conduct regression modeling or stratified analyses to adjust for potential confounding variables such as age, race, gender, comorbidities, or socioeconomic status. As a result, causal relationships cannot be inferred from our findings. Future research should incorporate multivariate statistical methods to better isolate the effects of telehealth utilization, medication access, and demographic factors on glycemic outcomes. Additionally, longitudinal analyses could help determine whether observed trends persist beyond the pandemic period.

Although the case series analysis cannot overcome the inherent limitations of claims data, it was included to provide a more detailed view of individual patient experiences. Claims data lack important clinical context, such as medication adherence, lifestyle behaviors, and psychosocial stressors, all of which are essential to understanding glycemic control. While not designed to establish causality, the case series offers valuable descriptive insights into patterns of medication continuity and telehealth engagement. These findings help contextualize the broader trends observed in the cohort analysis and point to the need for more comprehensive data collection in future studies.

While we observed improved glycemic control during the COVID-19 era, attributing this outcome to patient adaptation—such as medication rationing or enhanced self-care—is speculative given the limitations of our dataset. Our data do not include direct measures of patient behavior, lifestyle changes, or psychosocial factors. As such, any interpretation regarding patient-driven adaptation should be considered a hypothesis rather than a conclusion. Future research should incorporate patient-reported outcomes and contextual variables to better understand the behavioral mechanisms underlying glycemic trends during healthcare disruptions.

The significant increase in telehealth utilization during the COVID-19 era likely supported continuity of care. Although our study did not directly assess the impact of telehealth on glycemic outcomes, its expanded use may have facilitated ongoing communication, medication management, and monitoring. However, we acknowledge that no causal relationship was established between telehealth engagement and improved A1c outcomes. The observed association may reflect care continuity rather than a direct therapeutic effect. To avoid overstating its impact, we frame telehealth as a potential facilitator of diabetes management during healthcare disruptions, rather than a proven driver of improved outcomes. This interpretation aligns with prior studies that have demonstrated improved glycemic control through virtual care [2,22] and highlighted the value of telemedicine in underserved populations [22]. Future research should use regression modeling or stratified analysis to examine the relationship between telehealth engagement and A1c outcomes. Importantly, glycemic control is multifactorial. While insulin use was a focus of the case series, other factors, such as adherence to oral medications, lifestyle changes, and social determinants of health, also play critical roles. Our findings should be interpreted within this broader context.

Looking ahead, our findings underscore the potential of integrated care models that combine telehealth with pharmacist-led interventions to support chronic disease management. The expanded use of telehealth during the COVID-19 pandemic likely contributed to maintaining continuity of care. Future research should explore how virtual visits and pharmacist-supported medication management can be optimized to improve outcomes, particularly among underserved populations who face persistent barriers to care.

While our analysis identified patterns in prescription continuity, it is important to clarify that these findings do not confirm actual medication adherence. Prescription records indicate that a medication was ordered, but they do not verify whether it was filled, picked up, or taken as directed. Therefore, our conclusions regarding medication access and glycemic control should be interpreted as reflecting continuity in prescribing rather than confirmed patient adherence.

It is also important to note that our dataset did not distinguish between phone and video telehealth visits. These modalities may differ in their effectiveness for diabetes management, particularly in terms of patient-provider interaction, clinical assessment, and engagement. Future studies should explore whether the type of telehealth visit influences outcomes such as medication adherence, glycemic control, and care continuity.

While our case series suggested that patients with improved A1c often had more frequent telehealth visits, this descriptive analysis does not establish causality. Multivariate analyses are needed to account for potential confounders such as comorbidities, medication adherence, and socioeconomic factors.

The case series also provided insight into individual-level medication patterns. Patients who maintained or improved A1c control typically had uninterrupted access to both insulin and non-insulin therapies. These findings underscore the importance of ensuring consistent access to medications during healthcare disruptions. A 2011 study noted that insulin regimen selection is influenced by factors such as patient preferences, clinic resources, cost, and prescribing patterns [23]. As type 2 diabetes progresses, early initiation of insulin may be necessary when oral therapies are insufficient [24]. However, some patients may resist insulin initiation, preferring oral medications alone [24]. These individual-level insights into medication access and prescribing decisions provide important context for understanding broader patterns in diabetes care.

Notably, nearly two-thirds of the study population were Medicaid beneficiaries, highlighting the relevance of telehealth in supporting chronic disease management among low-income patients. As Medicaid policies continue to evolve, particularly regarding reimbursement for virtual care, telehealth may offer a critical mechanism for improving access, continuity, and equity in diabetes care. However, disparities in digital literacy, device access, and broadband connectivity may limit its effectiveness. Future research should examine how telehealth implementation within Medicaid programs can be optimized to support sustained engagement and improved outcomes in vulnerable populations.

This study also underscores the resilience of patients and the adaptability of outpatient care systems during a public health crisis. At the same time, it reveals critical vulnerabilities, especially in medication access, that must be addressed to ensure effective chronic disease management during future emergencies.

### Limitations

This study has several limitations that should be considered when interpreting the findings. First, the use of prescription records does not fully capture patient adherence. These records indicate that a medication was prescribed, but not whether it was filled, picked up, or taken as directed. As a result, our ability to assess true adherence and continuity of therapy is limited. Additionally, patients may have obtained medications from other providers, pharmacies, or emergency sources not reflected in the dataset, particularly during the pandemic, when care fragmentation was more likely. These factors introduce uncertainty into the interpretation of prescription gaps and their relationship to glycemic control. Some prescriptions may have ended due to side effects or clinical decisions, rather than being “lost.” Others may have been supplemented by patients using previously stored medications or rationing supplies to maintain adherence.

The relatively small sample size limits the generalizability of our findings to other outpatient settings. Moreover, patients included in the study were required to have A1c values in both time periods, which may have introduced selection bias by excluding individuals with less consistent access to care.

Our definitions of medication adherence and telehealth engagement were based on available administrative data and may not fully reflect patient behavior or clinical intent. The retrospective design also limits our ability to draw causal inferences.

Importantly, this study did not examine the influence of demographic factors such as age, race, or gender on outcomes. Future research should incorporate regression analyses to adjust for these potential confounders and explore how demographic and social determinants of health may influence diabetes management during public health emergencies.

Additionally, our analysis of prescription trends was based on total prescription counts rather than patient-level data. This approach may overstate changes in prescribing patterns if a small number of patients experienced multiple therapy changes. Future studies should consider analyzing prescriptions at the individual patient level to more accurately assess the distribution and impact of medication access changes.

To strengthen future research on medication adherence and continuity of care, we recommend incorporating pharmacy fill data or patient-reported adherence measures. These data sources would provide a more accurate assessment of whether prescribed medications were actually obtained and used as directed, thereby enhancing the validity of conclusions regarding the relationship between medication access and glycemic control.

Finally, the use of a single HbA1c threshold (<8.0%) to define glycemic control may oversimplify the clinical complexity of diabetes management. While this cut-off aligns with guidance from the American College of Physicians for outpatient care, it may not reflect optimal targets for all patients, particularly younger or healthier individuals for whom more stringent goals (e.g., <7.0%) may be appropriate. This approach also does not account for individualized treatment plans based on age, comorbidities, hypoglycemia risk, or provider judgment. Moreover, categorizing patients solely based on crossing the 8.0% threshold may obscure clinically meaningful changes in A1c. For example, a reduction from 9.5% to 8.2% would not be classified as “improved,” while a marginal change from 8.1% to 7.9% would. Future research should consider stratified analyses or continuous measures of A1c change to better capture the magnitude and clinical relevance of glycemic shifts.

## 5. Conclusions

This study found that glycemic control, as measured by A1c, significantly improved from the pre-COVID-19 period to the COVID-19 era, despite a reduction in prescriptions for diabetes medications and supplies. While telehealth utilization increased substantially during the pandemic, its direct impact on A1c outcomes was not assessed and remains an important area for future research.

The observed improvement in glycemic control should be interpreted with caution. It is possible that unmeasured confounding factors, such as changes in diet, physical activity, stress, or socioeconomic conditions, contributed to these outcomes. The reduction in prescriptions may also reflect care fragmentation or shifts in provider behavior rather than patient adaptation. These alternative explanations highlight the need for more comprehensive data and robust analytical approaches in future studies.

The case series analysis highlighted the critical role of uninterrupted access to insulin and other diabetes medications in maintaining or improving glycemic control. Patients who experienced therapy continuity were more likely to achieve better outcomes, underscoring the importance of resilient care delivery systems during public health emergencies.

Overall, these findings emphasize the need for integrated care models that ensure medication access and leverage telehealth to support chronic disease management, particularly in underserved populations. Future efforts should focus on strengthening care continuity and addressing systemic barriers to optimize diabetes outcomes during times of crisis and beyond.

## Figures and Tables

**Table 1 healthcare-13-02372-t001:** Demographic Characteristics of Patients with Type 2 Diabetes from Pre-COVID-19 to COVID-19 Era.

Demographic	Total (%)	Pre-COVID-19 to COVID-19 Era A1C Decreased from >8% to ≤8% (64 mmol/mol)	Pre-COVID-19 to COVID-19 Era A1C Increased from ≤8% to >8% (64 mmol/mol)	Pre-COVID-19 to COVID-19 Era A1C Remained ≤8% (64 mmol/mol)	Pre-COVID-19 to COVID-19 Era A1C Remained >8% (64 mmol/mol)
	n (%)	n (%)	n (%)	n (%)	n (%)
**Sex**					
Male	154 (39.8)	21 (39.6%)	12 (42.9%)	97 (40.8%)	24 (35.3%)
Female	233 (60.2)	32 (60.4%)	16 (57.1%)	141 (59.2%)	44 (64.7%)
**Race**					
American Indian or Alaska Native	2 (0.5)	0 (0.0%)	0 (0.0%)	2 (0.8%)	0 (0.0%)
Asian	10 (2.6)	2 (3.8%)	1 (3.6%)	6 (2.5%)	1 (1.5%)
Black or African American	143 (37.0)	22 (41.5%)	12 (42.9%)	87 (36.6%)	22 (32.4%)
Native Hawaiian or Other Pacific Islander	1 (0.3)	0 (0.0%)	0 (0.0%)	1 (0.4%)	0 (0.0%)
White	206 (53.2)	26 (49.1%)	14 (50.0%)	128 (53.8%)	38 (55.9%)
Unknown	25 (6.5)	3 (5.7%)	1 (3.6%)	14 (5.9%)	7 (10.3%)
**Ethnicity**					
Hispanic or Latino	93 (24.0)	10 (18.9%)	4 (14.3%)	59 (24.8%)	20 (29.4%)
Not Hispanic or Latino	192 (49.6)	27 (50.9%)	13 (46.4%)	123 (51.7%)	29 (42.6%)
Unknown	102 (26.4)	16 (30.2%)	11 (39.3%)	56 (23.5%)	19 (27.9%)
**Health Insurance**					
Medicaid	242 (62.5)	31 (58.5%)	19 (67.9%)	144 (60.5%)	48 (70.6%)
Medicare	96 (24.8)	15 (28.3%)	4 (14.3%)	65 (27.3%)	12 (17.6%)
Private Insurance	49 (12.7)	7 (13.2%)	5 (17.9%)	29 (12.2%)	8 (11.8%)
**Primary Language**					
English	331 (85.5)	45 (84.9%)	24 (85.7%)	210 (88.2%)	52 (76.5%)
Spanish	44 (11.4)	5 (9.4%)	2 (7.1%)	24 (10.1%)	13 (19.1%)
Others	8 (2.1)	3 (5.7%)	1 (3.6%)	2 (0.8%)	2 (2.9%)
Declined to specify	4 (1.0)	0 (0.0%)	1 (3.6%)	2 (0.8%)	1 (1.5%)

**Table 2 healthcare-13-02372-t002:** Comparison of A1c Control and Mean A1c Values Between Pre-COVID-19 and COVID-19 Eras.

Cohort	Number with A1c <= 8.0% (64 mmol/mol) n (%)	Number with A1c > 8.0% (64 mmol/mol) n (%)	Last A1c in the Defined Period (Mean, SD)
pre-COVID-19	266 (68.7)	121 (31.3)	7.66 ± 1.82
COVID-19 era	291 (75.2)	96 (24.8)	7.32 ± 1.74

**Table 3 healthcare-13-02372-t003:** Comparison of Diabetes-Related Prescriptions and Telehealth Utilization Between Pre-COVID-19 and COVID-19 Eras.

Cohort	Number Diabetes Medication Prescriptions *	Number Diabetes Supply Prescriptions	Number Insulin Prescriptions	Number Utilizing Telehealth Services *
pre-COVID-19	1293	889	528	3
COVID-19 era	1042	810	466	354

* *p* < 0.05.

**Table 4 healthcare-13-02372-t004:** Case Profiles of Patients with Changes in A1c Control Status, Insulin and Medication Use, and Telehealth Engagement During the COVID-19 Era.

Cases	Insulin Use Patterns	Prescription Pattern Description *	A1c Change Pre- to COVID-19 Era	Number of Telehealth Visit(s)
Uncontrolled-Patient Case 1	Insulin use only.	Insulin Lost long-acting and short-acting insulin during COVID-19 era. Other DM medications None.	A1c = 7.6 to 8.5 (60 to 69 mmol/mol)	1
Uncontrolled-Patient Case 2	Insulin use and other DM medication use.	Insulin Lost long-acting and short-acting insulin prescriptions during COVID-19 era. Other DM medications Lost one prescription during COVID-19 era that started in pre-COVID-19 era. Kept one prescription during COVID-19 era that started in pre-COVID-19 era. Initiated and stopped one prescription within COVID-19 era.	A1c = 7.5 to 8.7 (58 to 72 mmol/mol)	7
Uncontrolled-Patient Case 3	No insulin use but other DM medication use.	Insulin None. Other DM medications Kept one prescription through COVID-19 era.	A1c = 6.6 to 8.4 (49 to 68 mmol/mol)	4
Uncontrolled-Patient Case 4	Insulin use and other DM medication use.	Insulin Kept short-acting insulin through COVID-19 era. Lost long-acting insulin in COVID-19 era. Other DM medications Lost one prescription during COVID-19 era.	A1c = 5.6 to 14 (38 to 140 mmol/mol)	8
Uncontrolled-Patient Case 5	No insulin use, but other DM medication use.	Insulin None. Other DM medications One prescription, lost during COVID-19 era.	A1c = 7.7 to 10.1 (61 to 87 mmol/mol)	3
Controlled-Patient Case 6	Insulin use and other DM medication use.	Insulin Initiated short-acting insulin prescription during and kept through pre-COVID-19 era. Kept long-acting insulin through COVID-19 era. Other DM medications None before COVID-19 era. Initiated and lost one prescription during the COVID-19 era. Initiated and kept one prescription during the COVID-19 era.	A1c = 10.9 to 7.3 (96 to 56 mmol/mol)	7
Controlled-Patient Case 7	No insulin use but other DM medication use	Insulin None. Other DM medications Kept one prescription through COVID-19 era. Initiated and lost one prescription during COVID-19 era.	A1c = 10.8 to 8.0 (95 to 64 mmol/mol)	6
Controlled-Patient Case 8	Insulin use and other DM medications use	Insulin Kept short-acting and long-acting insulin prescriptions through COVID-19 era. Other DM medications Kept one prescription from pre-COVID-19 era through COVID-19 era.	A1c = 11.6 to 7.8 (103 to 62 mmol/mol)	19
Controlled-Patient Case 9	No insulin use but other DM medication use	Insulin None. Other DM medications Kept two prescriptions from pre-COVID-19 era through COVID-19 era.	A1c = 9.3 to 7.5 (78 to 59 mmol/mol)	3
Controlled-Patient Case 10	Insulin use and other DM medication use	Insulin Kept long-acting insulin prescription through COVID-19 era. Other DM medications Kept two prescriptions from pre-COVID-19 era through COVID-19 era.	A1c = 10.0 to 7.8 (86 to 62 mmol/mol)	6

* Lost = when a prescription ended without a new prescription of that same drug or drug within the same class following it immediately, creating a gap in care.

## Data Availability

The original contributions presented in this study are included in the article/Appendix A. Further inquiries can be directed to the corresponding author.

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
