# Peer review of "Impact of COVID-19 on A1c Management and Telehealth Use Among a Type 2 Diabetes Mellitus Population in the Outpatient Setting"

_healthcare, 2025, doi:10.3390/healthcare13182372_

Round 1

Reviewer 1 Report

Comments and Suggestions for Authors

The study provides good insight into diabetes care during the COVID-19 pandemic. The focus on COVID-19’s effect on diabetes care, particularly in underserved populations, is timely and important, however, multiple methodological and data limitations restrict the strength and generalizability of its findings. Please find below my comments:

1- The study relies on prescription records but cannot determine if medications were actually filled, picked up, or administered as directed. The lack of refill data means that true continuity of medication use, and actual adherence are not measured, weakening conclusions about the relationship between prescription access and glycemic control. For example, patients may have accessed medications from sources outside the study clinic, especially during pandemic disruptions, introducing error into assessments of medication continuity.

2- The counterintuitive improvement in HbA1c (despite reduced prescriptions) remains inadequately explained without behavioral data.

3- The HbA1C cut-off reflects a commonly accepted standard for older adults or patients with comorbidities where tighter glycemic targets may not be appropriate due to risks of hypoglycemia and other complications. However, other organizations and guidelines (American Diabetes Association) may set more stringent targets (<7.0%) for some populations, especially younger or healthier patients, and the appropriateness of the <8.0% threshold should be evaluated based on specific patient characteristics and clinical context. Moreover, using a single threshold of 8.0% for all patients oversimplifies glycemic control, when optimal targets should be individualized based on age, comorbidities, risk of hypoglycemia, and provider judgment.  For example, someone moving from 8.1% to 7.9% is classified as “improved” and counted in the positive outcomes (even if the change is marginal and potentially meaningless clinically). Similarly, a patient moving from 7.9% to 8.1% is classified as “worsened,” again based on a small difference. Additionally, the study focuses on crossing the threshold rather than the magnitude of improvement or worsening. For example, someone whose HbA1c decreased by 1% but still above 8.0% (e.g., 9.5% to 8.2%) would not be counted as “improved,” while a small decrease just under the cut-off would be.

4- The analysis of prescriptions (Table 3) counts the number of prescriptions, not the number of patients with prescriptions. This is a potential flaw as a single patient can have multiple prescriptions. A decline in total prescriptions could be driven by a small number of patients having multiple therapies discontinued, rather than a widespread trend.

5- The central finding (improved HbA1c with fewer prescriptions) is presented as a positive outcome of adaptation. However, an alternative and equally plausible interpretation is that clinical care was degraded. The reduction in prescriptions could indicate that providers were less aggressive in managing diabetes during the pandemic, perhaps due to fragmented care or a focus on acute issues. The improvement in HbA1c could then be attributed to unmeasured confounders (for example dietary changes during lockdown). The paper does not adequately consider this negative interpretation. Moreover, in the discussion, the authors attribute the improved control to patient "adaptation" (rationing medications, improved self-care) without any evidence to support this claim. The data shows only two things: HbA1c values and prescription records. Any story about why this happened is conjecture. Moreover, while telehealth was mentioned as a possible protective factor, no causal link was established. The conclusions about telehealth are overstated.

Author Response

Comment 1: The study relies on prescription records but cannot determine if medications were actually filled, picked up, or administered as directed. The lack of refill data means that true continuity of medication use, and actual adherence are not measured, weakening conclusions about the relationship between prescription access and glycemic control. For example, patients may have accessed medications from sources outside the study clinic, especially during pandemic disruptions, introducing error into assessments of medication continuity.

Response 1: We thank the reviewer for their thoughtful and constructive feedback. We appreciate the recognition of the study’s relevance and its focus on diabetes care in underserved populations during the COVID-19 pandemic. We agree that the use of prescription records alone limits our ability to assess true medication adherence. As noted in the manuscript’s limitations section, prescription data indicate that a medication was ordered but do not confirm whether it was filled, picked up, or taken as prescribed. We have clarified this point further in the revised discussion to emphasize that our findings reflect prescription continuity rather than confirmed adherence. We added the following: “While our analysis identified patterns in prescription continuity, it is important to clarify that these findings do not confirm actual medication adherence. Prescription records indicate that a medication was ordered, but they do not verify whether it was filled, picked up, or taken as directed. Therefore, our conclusions regarding medication access and glycemic control should be interpreted as reflecting continuity in prescribing rather than confirmed patient adherence.” Additionally, we acknowledge that patients may have accessed medications from external sources not captured in our dataset, particularly during the pandemic when care fragmentation was more likely. We have added language to the limitations section to reflect this concern and to suggest that future studies incorporate pharmacy fill data or patient-reported adherence measures to strengthen conclusions. Despite these limitations, we believe the observed associations between prescription continuity and A1c outcomes, especially in the case series, offer valuable insights into care patterns during a public health emergency. We have revised the manuscript to ensure that these findings are presented with appropriate caution and context.

Comment 2: The counterintuitive improvement in HbA1c (despite reduced prescriptions) remains inadequately explained without behavioral data.

Response 2: We appreciate the reviewer’s observation and agree that the improvement in HbA1c, despite a reduction in prescriptions, warrants further exploration. As noted in the revised discussion, our study did not include behavioral data such as changes in diet, physical activity, or stress levels, which may have influenced glycemic outcomes. We have now emphasized that this limitation restricts our ability to fully explain the observed trends. We have also added a recommendation for future research to incorporate behavioral and psychosocial data to better understand the mechanisms behind glycemic changes during healthcare disruptions.

Comment 3: The HbA1C cut-off reflects a commonly accepted standard for older adults or patients with comorbidities where tighter glycemic targets may not be appropriate due to risks of hypoglycemia and other complications. However, other organizations and guidelines (American Diabetes Association) may set more stringent targets (<7.0%) for some populations, especially younger or healthier patients, and the appropriateness of the <8.0% threshold should be evaluated based on specific patient characteristics and clinical context. Moreover, using a single threshold of 8.0% for all patients oversimplifies glycemic control, when optimal targets should be individualized based on age, comorbidities, risk of hypoglycemia, and provider judgment.  For example, someone moving from 8.1% to 7.9% is classified as “improved” and counted in the positive outcomes (even if the change is marginal and potentially meaningless clinically). Similarly, a patient moving from 7.9% to 8.1% is classified as “worsened,” again based on a small difference. Additionally, the study focuses on crossing the threshold rather than the magnitude of improvement or worsening. For example, someone whose HbA1c decreased by 1% but still above 8.0% (e.g., 9.5% to 8.2%) would not be counted as “improved,” while a small decrease just under the cut-off would be.

Response 3: We thank the reviewer for this insightful comment. We acknowledge that the use of a single HbA1c threshold (<8.0%) may oversimplify the complexity of glycemic control and does not account for individualized treatment goals based on patient age, comorbidities, and clinical context. The <8.0% cut-off was selected based on guidance from the American College of Physicians for outpatient diabetes management, particularly in populations with higher risk for hypoglycemia or limited access to care. However, we recognize that other organizations, such as the American Diabetes Association, recommend more stringent targets for certain subgroups. To address this concern, we have clarified in the revised manuscript that the <8.0% threshold was used as a pragmatic benchmark for population-level analysis, not as a universal clinical target. We have also added language to the limitations section acknowledging that this approach may obscure meaningful changes in glycemic control, such as improvements that do not cross the threshold or small fluctuations that may not be clinically significant. Furthermore, we now recommend that future studies consider stratified analyses or continuous measures of HbA1c change to better capture the magnitude and clinical relevance of glycemic shifts.

Comment 4: The analysis of prescriptions (Table 3) counts the number of prescriptions, not the number of patients with prescriptions. This is a potential flaw as a single patient can have multiple prescriptions. A decline in total prescriptions could be driven by a small number of patients having multiple therapies discontinued, rather than a widespread trend.

Response 4: We appreciate the reviewer’s observation. It is correct that Table 3 reflects the total number of prescriptions rather than the number of unique patients receiving prescriptions. We acknowledge that this approach may overrepresent changes in prescribing patterns, particularly if a small subset of patients experienced multiple therapy discontinuations. To address this limitation, we have clarified in the revised Methods and Limitations sections that prescription counts do not equate to patient-level prescribing trends. We have also added a recommendation for future research to analyze prescription data at the patient level to better assess the breadth and distribution of medication access changes across the population.

Comment 5: The central finding (improved HbA1c with fewer prescriptions) is presented as a positive outcome of adaptation. However, an alternative and equally plausible interpretation is that clinical care was degraded. The reduction in prescriptions could indicate that providers were less aggressive in managing diabetes during the pandemic, perhaps due to fragmented care or a focus on acute issues. The improvement in HbA1c could then be attributed to unmeasured confounders (for example dietary changes during lockdown). The paper does not adequately consider this negative interpretation. Moreover, in the discussion, the authors attribute the improved control to patient "adaptation" (rationing medications, improved self-care) without any evidence to support this claim. The data shows only two things: HbA1c values and prescription records. Any story about why this happened is conjecture. Moreover, while telehealth was mentioned as a possible protective factor, no causal link was established. The conclusions about telehealth are overstated.

Response 5: We appreciate the reviewer’s thoughtful critique and agree that alternative interpretations of our findings deserve further consideration. In response, we have revised the Conclusion section to more explicitly acknowledge the possibility that reduced prescription volume may reflect fragmented care or provider-level changes in treatment aggressiveness during the pandemic. We also recognize that attributing improved glycemic control to patient “adaptation” (e.g., rationing, self-care) is speculative given the limitations of our dataset. To address this, we have softened language around patient behaviors and clarified that these interpretations are hypotheses rather than conclusions. We now emphasize the need for future research to directly measure behavioral and contextual factors. Regarding telehealth, we have revised the manuscript to avoid overstating its impact. While our data show increased telehealth utilization, we acknowledge that no causal relationship with glycemic outcomes was established. We now frame telehealth as a potential facilitator of care continuity rather than a proven driver of improved outcomes.

Reviewer 2 Report

Comments and Suggestions for Authors

Thank you for submitting the manuscript. Below are some recommendations to strengthen the manuscript.

Introduction: It is unusual to find in-text citations in abstract unless they were intended for the first paragraph in the introduction section.

Methods: How many participants were screened for eligibility and how many were included? This information is only provided in the results section. Telehealth visits were defined as virtual/phone visit, explicitly state if virtual also included video calls? If data was available, would phone and video call affect the outcomes measures?

Statistical analyses: Including what was not done and recommendations for future studies should be included in the discussion section.

Results: How were the 10 patients randomly selected for Table 4?

Discussion: The last portion of the limitation section, lines 267-280 are redundant as the authors have addressed most of them in the introduction and/or methods section. My recommendation would be to remove it from introduction/methods and include it only in the discussion section. Nearly 2/3rd of the participants were Medicaid patients, what implications does telehealth have on chronic disease management among Medicaid patients given the changing landscape of Medicaid. 

Author Response

Comment 1: Introduction: It is unusual to find in-text citations in abstract unless they were intended for the first paragraph in the introduction section.

Response 1: We appreciate the reviewer’s observation regarding the presence of in-text citations in the abstract. These citations were inadvertently included and were intended for the introductory paragraph of the manuscript. We have revised the abstract to remove all in-text citations.

Comment 2:
Methods: How many participants were screened for eligibility and how many were included? This information is only provided in the results section. Telehealth visits were defined as virtual/phone visit, explicitly state if virtual also included video calls? If data was available, would phone and video call affect the outcomes measures?

Response 2: We thank the reviewer for highlighting the need for greater clarity in the Methods section. In response, we have added a statement to the Methods section specifying the number of patients screened and the number included in the final analysis to improve transparency and reproducibility. Additionally, we have clarified that telehealth visits included both phone and video calls, as defined by the electronic health record coding. While our dataset did not differentiate between phone and video modalities, we acknowledge that these may have different impacts on care delivery and outcomes. We have added a statement to the Methods and Discussion sections to reflect this limitation and suggest it as an area for future research.

Comment 3: Statistical analyses: Including what was not done and recommendations for future studies should be included in the discussion section.

Response 3: We appreciate the reviewer’s suggestion to expand the Discussion section by including what statistical analyses were not performed and recommendations for future studies. In response, we have added a paragraph to the Discussion that outlines the limitations of our statistical approach, including the absence of regression modeling and stratified analyses. We also provide recommendations for future research to incorporate multivariate methods to better account for confounding variables and explore causal relationships.

Comment 4: Results: How were the 10 patients randomly selected for Table 4?

Response 4: We thank the reviewer for this important question. The 10 patient cases presented in Table 4 were randomly selected from the pool of patients whose A1c status changed between the pre-COVID-19 and COVID-19 era. Specifically, five patients whose A1c worsened and five whose A1c improved were selected using a simple random sampling method from each respective group. We have added a clarifying statement to the Methods section 2.4 to describe this selection process.

Comment 5: Discussion: The last portion of the limitation section, lines 267-280 are redundant as the authors have addressed most of them in the introduction and/or methods section. My recommendation would be to remove it from introduction/methods and include it only in the discussion section. Nearly 2/3rd of the participants were Medicaid patients, what implications does telehealth have on chronic disease management among Medicaid patients given the changing landscape of Medicaid. 

Response 5: We appreciate the reviewer’s feedback regarding redundancy in the limitations section. In response, we have consolidated the discussion of limitations by removing overlapping content. Additionally, we acknowledge the importance of exploring the implications of telehealth for Medicaid patients, especially given that nearly two-thirds of our study population were Medicaid beneficiaries. We have added a paragraph to the Discussion section addressing how telehealth may influence chronic disease management in this population, particularly in light of evolving Medicaid policies and coverage models.

Round 2

Reviewer 1 Report

Comments and Suggestions for Authors

The authors have addressed my comments.